# Graves’ Disease during Immune Checkpoint Inhibitor Therapy (A Case Series and Literature Review)

**DOI:** 10.3390/cancers13081944

**Published:** 2021-04-17

**Authors:** Mathilde Peiffert, Christine Cugnet-Anceau, Stephane Dalle, Karim Chikh, Souad Assaad, Emmanuel Disse, Gérald Raverot, Françoise Borson-Chazot, Juliette Abeillon-du Payrat

**Affiliations:** 1Faculté de Médecine, Université Lyon 1, 69008 Lyon, France; stephane.dalle@chu-lyon.fr (S.D.); karim.chikh@chu-lyon.fr (K.C.); emmanuel.disse@chu-lyon.fr (E.D.); gerald.raverot@chu-lyon.fr (G.R.); francoise.borson-chazot@chu-lyon.fr (F.B.-C.); 2Fédération d’Endocrinologie, Hôpital Louis Pradel, Groupement Hospitalier Est, Hospices Civils de Lyon, 69500 Bron, France; 3Service d’Endocrinologie-Diabète-Nutrition, Hôpital Lyon Sud, Hospices Civils de Lyon, 69310 Pierre-Bénite, France; christine.cugnet-anceau@chu-lyon.fr; 4ImmuCare, Institut de Cancérologie, Hospices Civils de Lyon, 69002 Lyon, France; 5Service de Dermatologie, Hôpital Lyon Sud, Hospices Civils de Lyon, 69310 Pierre-Bénite, France; 6Centre de Biologie Sud, Hôpital Lyon Sud, Hospices Civils de Lyon, 69310 Pierre-Bénite, France; 7Tox’imm, Centre Léon Bérard, 69008 Lyon, France; souad.assaad@lyon.unicancer.fr; 8Service d’Hématologie et Médecine Interne, Centre Léon Berard, 69008 Lyon, France; 9INSERM U1060, INRA 1397, INSA Lyon, Centre de Recherche en Nutrition Humaine Rhône-Alpes (CRNH RA), CarMeN Laboratory, 69310 Pierre-Bénite, France; 10INSERM U1052, CNRS, UMR5286, Cancer Research Center of Lyon, 69008 Lyon, France

**Keywords:** Graves’ disease, immune checkpoint inhibitors, immune-related adverse event, endocrine toxicities, thyroid dysfunction

## Abstract

**Simple Summary:**

Immune checkpoint inhibitor (ICPi)-induced thyroid dysfunction is a frequent immune-related adverse event (irAE). ICPi-induced thyrotoxicosis is usually the first stage of a biphasic thyroiditis with secondary hypothyroidism, whereas ICPi-induced Graves’ disease (GD), due to the stimulating activity of TSH-receptor autoantibodies, is extremely rare. The aim of this study was to describe the characteristics and evolution of GD during ICPi therapy. We showed that in five patients with induced GD, two patients evolved into classical GD and the three other patients evolved as thyroiditis with short-term thyrotoxicosis and secondary long-term hypothyroidism, with the initial scintigraphic appearance seeming to predict their evolution. Three other patients had a diagnosis of GD before ICPi treatment: all evolved towards definitive hypothyroidism during treatment, without the appearance of irAE. None of the eight patients developed severe hyperthyroidism with life-threatening symptoms, nor significant Graves’ orbitopathy. Use of ICPis in this population with induced or pre-existing autoimmune GD disease therefore appears to be safe.

**Abstract:**

Thyrotoxicosis is an adverse event associated with immune checkpoint inhibitors (ICPis) that occurs in 0.6 to 3.2% of treated patients, depending on ICPi class. Presentation usually consists of a biphasic thyroiditis with transient thyrotoxicosis and secondary hypothyroidism. ICPi-induced Graves’ disease (GD), due to the stimulating activity of TSH-receptor autoantibodies (TRAb), is extremely rare. The aim of this retrospective study was to describe the characteristics and evolution of GD during ICPi therapy. Five among 243 patients followed for ICPi-induced thyrotoxicosis showed TRAb positivity (2% of the cohort). GD occurred quickly after initiation of ICPis; its course was typical for two patients, with prolonged requirement for antithyroid drug treatment (ATD). The three other patients experienced biphasic thyroiditis with secondary hypothyroidism requiring long-term substitution. Three other patients had a diagnosis of GD before starting ICPis; they evolved toward hypothyroidism with early cessation of ATD and long-term substitution treatment during ICPi treatment. None developed significant Graves’ orbitopathy. ICPi treatment was not interrupted for thyroid dysfunction. In conclusion, GD is a rare, immune-related adverse event of ICPis with an unusual course and frequent evolution to biphasic thyroiditis. In the case of ICPi-induced thyrotoxicosis in the presence of TRAb, observing the spontaneous evolution and performing a scintigraphy are useful before starting ATD treatment. Pre-existing GD is not exacerbated by ICPis and tends to evolve towards hypothyroidism. ICPi treatment can be maintained with adequate biochemical surveillance.

## 1. Introduction

Immunotherapies are recent anti-cancer treatments that have been developed over the last fifteen years and which now play a major role in the treatment of several cancers, including melanoma, lung, and kidney. Their mechanism of action is via the reactivation of T lymphocytes through targeting immune checkpoints that are used by cancer cells to cause immunosuppression. Immune checkpoint inhibitors (ICPis) target different immune checkpoints and consist of monoclonal antibodies directed against cytotoxic T-lymphocytic-associated antigen-4 (anti-CTLA-4 antibodies: Ipilimumab, Tremelimumab), programmed cell death 1 protein (anti-PD-1 antibodies: Nivolumab, Pemprolizumab and Cemiplimab), and programmed cell death ligand-1 (anti-PD-L1 antibodies: Atezolizumab, Durvalumab and Avelumab). Other agents in this class of drugs are also currently under clinical trials. The range of indications for these treatments has been broadening each year [1].

However, by inhibiting negative regulators of adaptive immunity, ICPis may also trigger immune-related adverse events (irAE) involving different organs, with a frequency of 14% of grade 3 and 4 irAE observed with anti-PD-(L)1 treatment, 34% with anti-CTLA-4 treatment, and 55% with combined immunotherapy [2].

The endocrine irAEs involve several endocrine glands (pituitary, thyroid, adrenals, and pancreas) [3]. Anti-CTLA-4 preferentially exhibited pituitary toxicity with hypophysitis (up to 10%), whereas anti-PD-(L)1 more frequently caused thyroid diseases, estimated at 1 to 10%, and up to 50% when subclinical dysthyroidism and systematic screening are included [4,5,6,7,8]. Frequency of frank hypothyroidism is estimated to be 6.6% with ICPis, whereas the reported incidence of hyperthyroidism is lower than that of hypothyroidism (3.2% of patients treated with anti-PD-1, 0.6% with anti-PD-L1 and 1.7% with anti-CTLA-4) [6,9]. The hyperthyroid phase typically evolves into hypothyroidism, which may explain many cases of hyperthyroidism being missed in clinical studies [10].

Thyroid dysfunctions (TDs) generally appear during the first weeks of ICPi treatment, although they may in some cases be delayed [7,11,12]. Pre-existing TD and combined or successive use of ICPis seem to multiply these thyroid side effects by 3–4-fold [4,5,6,11,12,13]. Most patients with hypothyroidism treated by Levothyroxine prior to ICPi treatment need to increase their dosage. A few patients with prior hypothyroidism developed hyperthyroidism after the initiation of ICPis [11]. A dose-effect is also suggested by a study that used a high dose of Ipilimumab (10 mg/kg), where 10% of patients developed dysthyroidism [14]. Thus, it is recommended that systematic pre-therapeutic evaluations of endocrine function are carried out, then close monitoring during the first months of treatment [15,16]. ICPi-induced thyrotoxicosis is generally due to destructive thyroiditis, with a first stage of hyperthyroidism which then resolves or evolves into secondary hypothyroidism (50–80%) [17,18,19].

Graves’ disease (GD) is an autoimmune disease (AID) due to the stimulating activity of TSH-receptor autoantibodies (TRAb). Its diagnosis is based on the association of thyrotoxicosis and the presence of TRAb positivity and/or orbitopathy. GD is unusual during ICPi treatment and has so far rarely been described in the literature.

The aim of this study was to describe the profile of thyrotoxicosis in the presence of TRAb in patients undergoing treatment with ICPis.

## 2. Materials and Methods

A descriptive cohort study with longitudinal follow-up was conducted, in the tertiary referral center of the University Hospital of Lyon (Hospices Civils de Lyon), France. Patients included were referred either by dermatologists, respiratory physicians, or oncologists for thyrotoxicosis during ICPi treatment, from December 2015 to December 2020, through the ImmuCare and Tox’imm networks (healthcare networks in Lyon including patients who have presented irAEs due to ICPis during treatment for cancer) [20].

We included adult patients presenting with thyrotoxicosis diagnosed at pre-therapeutic screening or during follow up for ICPi treatment for any cancer, and showing as positive for TRAbs.

The diagnosis of thyrotoxicosis was accepted in the case of low thyroid stimulating hormone (TSH) levels (normal values: 0.4–4 mUI/L) with concomitant high free thyroxine (FT4, normal values: 12–22 pmol/L) and/or high free triiodothyronine (FT3, normal values: 3.1–6.8 pmol/L) (chemiluminescence immunoassay (I2000; Abbott Laboratories, Abbott Park, IL, USA)).

TRAb levels were considered positive if above 1.6 UI/L (electrochemiluminescence, Cobas e411; Roche Diagnostics, Mannheim, Germany). In one case, a TRAb bioassay was performed to determine the stimulating activity. Thyroperoxidase autoantibodies (TPOAb) were considered positive if above 34 UI/mL (electrochemiluminescence immunoassay, Cobas e411; Roche Diagnostics, Mannheim, Germany).

When required, morphological assessment was based on cervical Ultrasound-Doppler examination and functional assessment by 99mTc uptake on thyroid scintigraphy.

Secondary hypothyroidism evolution was diagnosed in the case of high TSH with low FT4/FT3 in the absence of any thyroid treatment.

Patient follow-up was based on clinical examination and biochemical follow-up, the frequency of which was at the discretion of the endocrinologist.

Data were processed anonymously. In accordance with legislation in place at the time of the study, the study was declared to the data protection agency (Commission Nationale Informatique et Liberté; no. 18-297) and an information letter was sent to living patients; written informed consent was not required. The study was also approved by the Ethics Committee of the Hospices Civils de Lyon (no. 19-163.)

## 3. Results

### 3.1. Patients

A total of 243 patients (150 male and 93 female) were referred for thyrotoxicosis during ICPi treatment. Five patients developed induced-thyrotoxicosis with positive TRAb (2% of the cohort) and were included in the “ICPi-induced Graves’ disease” group. Additionally, three patients had been diagnosed for Graves’ disease prior to starting ICPis (TRAb-positive thyrotoxicosis), and these patients were included in the “Pre-therapeutic Graves’ disease” group (Figure 1).

### 3.2. ICPi-Induced Graves’ Disease

Five patients (numbers 1 to 5, Table 1.) were treated with anti-PD-1 monotherapy or with a combination of anti-PD-(L)1 and anti-CTLA4. Patients 4 and 5 were current smokers. None of the patients had a personal history of autoimmune disease nor of thyroid dysfunction; their family histories were unknown. All had normal pre-therapeutic thyroid function, and none had previously received an anti-cancer treatment that induced thyroid adverse events. For four patients, thyrotoxicosis was diagnosed quickly after ICPi initiation, while for patient 1 diagnosis was much later during the maintenance phase. None showed any clinical criteria for severe hyperthyroidism (cardiothyreosis), nor any alteration in general state requiring hospitalization, thanks to systematic biochemical screening of thyroid function. None of these patients developed significant Graves’ orbitopathy requiring specific treatment.

For patients 1 and 2, the course of the thyrotoxicosis was typical of a Graves’ hyperthyroidism (Figure 2A.): both were treated with an antithyroid drug (ATD) and TSH normalized quickly; TRAb normalization occurred after more than 6 months. Total duration of ATD treatment was about one year for patient 1, with a slow decrease in ATD treatment until spontaneous euthyroidism; patient 2 was still undergoing treatment with ATD after 6 months because early tapering of treatment resulted in early recurrence of hyperthyroidism. Both patients were treated with corticotherapy for an additional grade 3 irAE (for patient 1, subsequent autoimmune hepatitis, and for patient 2, consecutive autoimmune pneumopathy); these irAEs resulted in the withdrawal of ICPis.

Patients 3, 4 and 5 experienced transient thyrotoxicosis shortly after their first ICPi treatment (Figure 2B). Only patient 5 required ATD, but this was stopped early, and the two others only received symptomatic treatment (beta-blockers). In patients 3 and 5, thyrotoxicosis evolved to hypothyroidism after a few months, requiring long-term substitution of thyroid hormone. Patient 4 died soon after spontaneous evolution towards hypothyroidism (low FT3/FT4 and normalized TSH) one month after the discontinuation of ICPi for a grade 3 irAE (autoimmune hepatitis), without glucocorticoid treatment. No information was available regarding the length of time taken until these patients were negative for TRAbs.

None of these five patients with ICPi-induced GD had treatment interruption due to thyroid dysfunction.

### 3.3. Pre-Therapeutic Graves’ Disease

Three patients (numbers 6, 7 and 8, Table 2) were diagnosed with GD before starting ICPi treatment; all three received anti-PD-1 (1 Pembrolizumab, 2 Nivolumab).

For patients 6 and 7 (Figure 2C), GD was fortuitously diagnosed with thyrotoxicosis at pre-therapeutic screening (retrospectively in the case of patient 7 whose symptoms evolved over one month; he was also a current smoker). Patient 8 was treated for GD 1.5 years before ICPi treatment; TRAb levels were at the time six-fold greater than the normal upper limit, (ULN) and TPOAb were also positive. The patient was treated with Methimazole for one year until TRAbs tested negative. He was then diagnosed with a renal clear cell carcinoma, firstly treated with a tyrosine-kinase inhibitor (Pazopanib) for one year, with euthyroidism, and then with Nivolumab. GD recurrence was diagnosed during pre-ICPi treatment biochemical examination: TRAbs were higher than the first time, at 14 ULN, with moderate stimulating activity (324%).

All three of these patients were treated with ATD together with ICPis. In all patients, the development of secondary hypothyroidism allowed the early cessation of ATD treatment, whereas TRAbs remained positive. They all required long-term substitution treatment for persistent hypothyroidism, using high doses of Levothyroxine.

None of these patients developed significant Graves’ orbitopathy requiring specific treatment, nor developed irAE during ICPi treatment.

## 4. Discussion

ICPi-induced Graves’ disease is rare and has been seldom reported in the literature, with only a few case reports published. Although limited, our series is the largest and gives indications on the clinical characteristics and evolution of these patients and the safety of ICPi use in these cases.

GD is an autoimmune pathology, presenting with the association of thyrotoxicosis and TRAb positivity and/or orbitopathy. Functional and morphological examinations typically show high iodine/99mTc uptake on thyroid scintigraphy and a hypervascular pattern on Ultrasound-Doppler. Orbitopathy is associated in almost half of the cases. Specific treatment consists of ATD, usually maintained for one year, allowing a return to spontaneous euthyroidism. Recurrence occurs in 50% of cases, whereas 15% of patients show evolution to secondary hypothyroidism, particularly those that are TPOAb-positive [21,22,23,24,25].

Positive TRAbs were described in 2% of all patients referred for ICPi-induced thyrotoxicosis in our cohort, and this prevalence is probably an overestimate because some patients with mild thyrotoxicosis would certainly not have been referred to an endocrinologist. Patients were treated using different types of ICPi (anti-PD-1, anti-CTLA-4, and a combination of anti-CTLA-4 with anti-PD-(L)1). None had frank orbitopathy that required specific treatment, and no patients were referred for Graves’ orbitopathy without thyroxicosis. Thyrotoxicosis symptoms were generally modest, despite high hormones levels, maybe due to their early diagnosis thanks to systematic screening. The majority of induced GD appeared at the beginning of ICPi treatment, but there is a need for the prolonged monitoring of thyroid function because one patient developed GD much later, after 41 weeks of treatment.

Usually, the evolution of ICPi-induced thyrotoxicosis presents similar to destructive thyroiditis, with initial hyperthyroidism followed by resolution or evolution towards hypothyroidism in 50–80% of cases [17,18,19]. Complementary investigations are not useful, except in the case of severe thyrotoxicosis or an unfavorable evolution: TRAb assay, Ultrasound-Doppler, and iodine/99mTc scintigraphy should then be performed to discriminate with differential diagnoses [15]. In typical thyroiditis, thyroid Ultrasound-Doppler shows a heterogeneous hypoechoic parenchyma with low vascularity and low iodine/99mTc uptake on scintigraphy [7]. However, many patients have iterative scans using iodinated contrast agents which can interfere with iodine uptake in scintigraphy. Histological and morphological analysis can demonstrate different pathophysiological mechanisms between autoimmune and ICPi-induced thyrotoxicosis [26,27].

In our series, 75% of patients with GD during ICPi treatment evolved towards hypothyroidism, which is more similar to the evolution of biphasic thyroiditis than that of classical GD. Scintigraphy, when performed, was useful in predicting the evolution, showing high fixation in the case of classical GD, and no fixation in the case of thyroiditis.

Despite the evolution to hypothyroidism, TRAb remained positive for a long period of time. Evolution towards hypothyroidism in some patients could be due to a conversion of TRAbs from stimulating to blocking antibodies, or to a thyroiditis added to Graves’ disease; unfortunately, we could not perform TRAb bioassays to conclude. Similar observations were made in Alemtuzumab use for multiple sclerosis in the 1990s: during the reconstitution of autoimmunity, a significant proportion of patients developed GD, with an unusual evolution towards spontaneous euthyroidism or hypothyroidism in 40% of the patients, despite TRAb positivity. [28].

In the literature, initially only induced Graves’ orbitopathy (GO) without thyrotoxicosis were described in patients undergoing Ipilimumab (anti-CTLA-4) treatment [29,30,31]. Polymorphism of CTLA-4 is known to increase the risk of autoimmune susceptibility including Graves’ disease, but how it interacts with ICPis is unclear [32,33,34]. Nevertheless, a small number of more recent case reports have described Graves’ disease associated with anti-PD-1 ICPi treatment. The later arrival of their use in oncology could explain this time lag. To the best of our knowledge at the time of writing, only nine cases of ICPi-induced Graves’ hyperthyroidism and seven cases of induced Graves’ orbitopathy have been described in the literature, associated with anti-PD-1 or anti-CTLA-4 treatment or their combination (Table 3), with only one patient presenting with both thyrotoxicosis and orbitopathy [19,29,30,31,35,36,37,38,39,40,41,42,43,44]. All eight cases of induced GD without orbitopathy showed TRAb positivity and/or high iodine uptake on scintigraphy. These patients were treated using ATD, and only one evolved towards spontaneous hypothyroidism. The only case with an association of induced hyperthyroidism and orbitopathy was reported by Sagiv et al. and was treated with glucocorticoids, and hyperthyroidism resolved spontaneously [42]. In our series, only 40% of induced GD evolved as classical GD would, while 60% showed spontaneous hypothyroidism, highlighting a heterogeneous profile.

Among the six case reports with Graves’ orbitopathy but without hyperthyroidism, all were treated with glucocorticoids for eye inflammation, and none presented with thyrotoxicosis nor showed evolution to hypothyroidism (Table 3). Of these six patients only two showed TRAb positivity, and we could reconsider the GO diagnosis in the four others because it may represent ICPi-induced orbitopathy rather than Graves’ disease [45].

To the best of our knowledge, there is only one reported case of pre-existing Graves’ hyperthyroidism before ICPi treatment [42]. This patient had a history of GD one year before immunotherapy; he commenced Nivolumab in a euthyroid state but experienced a recurrence of thyrotoxicosis with GO that evolved to hypothyroidism, despite persistent positive TRAbs. Our three patients with pre-existing GD showed the same course towards hypothyroidism during ICPi treatment, but did not develop GO.

In our case series of ICPi-induced GD, patients with high TRAb or high FT4 at diagnosis later showed a thyroiditis-like evolution, and not all patients with TPOAb developed hypothyroidism: neither TPOAb, nor TRAb value, nor FT4 initial value seems to be predictive of classical GD or thyroiditis-like evolution.

In the literature, female gender is described as being a risk factor for ICPi-induced TD, although this is less pronounced than in spontaneous forms [7,46,47] and it is not described in all studies [11,48]. In our entire cohort of 243 patients with ICPi-induced thyrotoxicosis, males were over-represented (60%). The literature review of ICPi-induced Graves’ hyperthyroidism (Table 3) shows a higher proportion of males than females in the cases previously reported (7/8), which is unexpected, because GD is usually a predominantly female condition. This was not the case in our cohort of GD (females were over-represented at 60%), likely due to the small numbers.

Clinical trials of ICPis excluded patients with pre-existing active autoimmune diseases because of concerns of a high susceptibility to severe irAEs [49,50]. However, a growing body of evidence indicates that ICPis may be safe and effective in these patients [51,52]. A prospective study of 45 patients has shown that overall survival time and objective response rates did not differ between patients with a pre-existing AID and patients without a pre-existing AID when treated with anti-PD-1 antibodies. Nonetheless, a pre-existing AID was associated with a significantly increased risk of irAEs [53]. Although irAEs may resemble AID, their pathophysiology remains poorly understood. One important explanation may be that following immune checkpoint inhibition, T cells and other immune cells are activated, resulting in the production of pro-inflammatory cytokines which may lead to off-target inflammation and autoimmunity [51]. Among the three cases of pre-existing GD in our cohort, two were diagnosed at their regular check-up just before commencement of ICPi treatment. Our results are reassuring on their evolution when treated with ICPis, with none developing Graves’ orbitopathy, severe thyrotoxicosis, nor other irAE during treatment, suggesting the safety of ICPis in this autoimmune context. Nevertheless, close monitoring of thyroid function is required because these patients may progress to secondary hypothyroidism.

Two retrospective and two prospective studies reported that positive TPOAb and/or thyroglobulin autoantibodies (TgAb) are associated with a 40 to 70% risk of hypothyroidism during treatment, and that patients who had ICPi-induced hypothyroidism had positive anti-thyroid autoantibodies in 75 to 100% of cases [54,55,56,57]. These elements suggest that ICPI-induced thyroid dysfunction may have a pathogenesis similar to spontaneous autoimmune hypothyroidism [16]. Guidelines recommend the evaluation of thyroid function before commencing ICPi treatment, with monitoring of thyroid function during the first months of treatment [15,16]. Even closer monitoring is needed if patients have pre-existing auto-immune thyroid, but systematic screening of thyroid antibodies is not recommended. Note that in the Osorio study [56], survival was better if patients developed ICPi-induced thyroid dysfunction, as suggested by others if there was an occurrence of irAEs [58,59,60].

Corticosteroid therapy may influence the course of ICPi-induced thyroid dysfunction [57,61]. Kimbara et al. noted that of 20 patients with induced thyrotoxicosis, five of the six who did not develop secondary hypothyroidism had received corticosteroid therapy for another reason [57]. This could also explain the evolution of our two patients who presented with induced GD showing classical evolution without hypothyroidism, because they had received corticosteroids for another irAE. A recent review of the literature is reassuring on the use of corticosteroids for irAE management, finding that it seems not to reduce overall survival in cancer patients treated with ICPis [62,63] and may be safely administered without compromising outcomes [16,64]. Conversely, a detrimental effect on survival has been suggested when used for cancer-related symptoms or for the management of a pre-existing AID, because high-dose steroids can theoretically reverse the anti-cancer benefit of ICPi treatment [51,62].

ICPi-induced thyrotoxicosis, when moderate, is usually controlled with symptomatic treatment. ESMO guidelines recommend temporarily suspending ICPi treatment in the case of severe thyrotoxicosis, which necessitates treatment with corticosteroids or ATD [16]. ICPi treatment can be re-initiated later with the agreement of the oncologist, and thyrotoxicosis should not be a reason for the definitive cessation of ICPis. The presence of an ICPi-induced endocrinopathy must not contraindicate the use of another anti-cancer therapy, including those of the same class [15]. In severe Graves’ orbitopathy, ICPis should be stopped in the case of sight-threatening prognosis. Published French guidelines recommend that the use of ATD should be reserved for hyperthyroidism with TRAb positivity or high iodine uptake on scintigraphy proving a primary thyroid pathology [15]. In the case of thyroiditis, ATD is ineffective because of the cytolytic mechanism underlying thyrotoxicosis. Our study shows that despite TRAb positivity, ATD is not always required because about one-half of the patients will spontaneously evolve towards hypothyroidism.

## 5. Conclusions

ICPi-induced Graves’ disease is very rare and represents about 2% of ICPi-induced thyrotoxicosis, although it is likely that this rate will be increased with the use of combinations of ICPis. TRAb-positivity is not predictive of the evolution of hyperthyroidism, and patients may show evolution similar to classic Graves’ disease, with long antithyroid drug therapy, or may present a classic pattern of thyroiditis which rapidly evolves to hypothyroidism. Evaluation of pre-therapeutic thyroid function is necessary to identify at-risk populations, and biochemical follow-up is recommended throughout ICPi treatment to facilitate the early diagnosis of thyroid dysfunction, particularly because the clinical signs are very non-specific in this neoplastic context. Testing for TRAb-positivity should be widely prescribed, especially in the absence of a rapid improvement of thyroid function. If patients are TRAb-positive, our work would favor carrying out thyroid scintigraphy to discriminate between patients whose thyrotoxicosis will quickly resolve and those who will require specific treatment. ICPi treatment in patients with pre-existing GD showed no exacerbation as a result of treatment and did not develop irAE. Lastly, GD should not be a contra-indication for ICPi treatment, subject to close biochemical monitoring and awareness of the risk of evolution toward hypothyroidism.

## Figures and Tables

**Figure 1 cancers-13-01944-f001:**
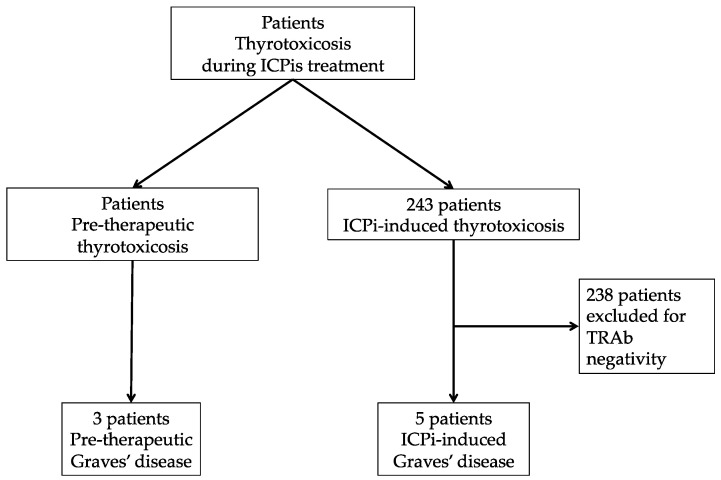
Flow chart. ICPis: immune checkpoint inhibitors; TRAb: TSH-receptor autoantibodies.

**Figure 2 cancers-13-01944-f002:**
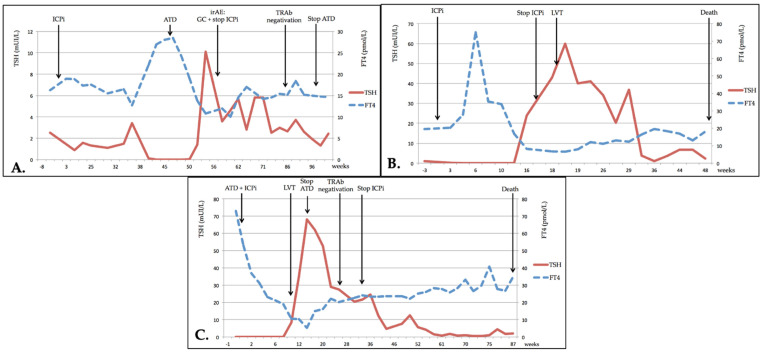
Evolution of thyroid dysfunction. (**A**) Patient no. 1; (**B**) Patient no. 3; (**C**) Patient no. 6. TSH: thyroid stimulating hormone; FT4: free thyroxine; ICPi: immune checkpoint inhibitor; ATD: antithyroid drug therapy; irAE: immune-related adverse event; GC: glucocorticoids; TRAb: TSH-receptor autoantibodies; LVT: levothyroxine.

**Table 1 cancers-13-01944-t001:** ICPi-induced Graves’ disease.

No.	Sex Age	Indication	ICPi (Name) and Dose	Cycle/Time Since ICPi Initiation	Total ICPi Cycles Number	Other irAE (Time)	Maximum FT4/FT3 (in ULN)	TRAb at Diagnosis (in ULN)	Iodine 99mTc Uptake/ Hyper-Vascular Doppler	TPOAb	ATD (Duration in Weeks)	Corticosteroids (Time of Introduction)	Status at Last Follow-Up	Treatment at Last Follow-Up (Time of Introduction)	Total Follow-Up (Weeks)
1	F 69	Serous ovarian carcinoma	anti-PD-1 (Pembrolizumab) 200 mg/3 weeks	IX/41 weeks	XIV	G3 (week 15)	1.3/1.3	2.7	High/na	Negative	Carbimazole (45)	Yes * (week 15)	Euthyroidism	0	57
2	F 60	Endometrial adenocarcinoma	anti-CTLA-4 and anti-PD-L1 (Durvalumab and Tremelimumab) 1500 mg and 76 mg	I/2 weeks	I	G3 (week 2)	2.5/1.8	5.2	na/No	Positive	Carbimazole (>29 **)	Yes * (week 2)	Hyperthyroidism	Carbimazole 1.25 mg/day (week 1)	30
3	M 60	Small cell bronchial carcinoma	anti-PD-1 (Pembrolizumab) 200 mg/3 weeks	I/3 weeks	VI	No	3.4/2	3.5	na/na	Positive	No	No	Hypothyroidism	Levothyroxine 150 µg/day (week 17)	48
4	F 44	Melanoma	anti-CTLA-4 and anti-PD-1 (Ipilimumab and Nivolumab) 3 and 1 mg/kg/3 weeks	I/2 weeks	II	G3 (week 5)	1.9/na	4.1	Low/No	na	No	No	DCD ***	0 ***	13
5	M 76	Epidermoid lung carcinoma	anti-PD-1 (Nivolumab) 3 mg/kg/2 weeks	II/4 weeks	XXXVIII	No	2.4/1.3	48	na/No	Negative	Carbimazole (10)	No	Hypothyroidism	Levothyroxine 150 µg/day (week 16)	143

Time is expressed in weeks, since the beginning of thyrotoxicosis (except where otherwise specified). FT4, FT3 and TRAb are expressed as number of times greater than normal levels. Normal values: FT4 12–22 pmol/L, FT3: 3.1–6.8 pmol/L, TRAb < 1.6 U/L. ICPi: immune checkpoint inhibitor; irAE: immune-related adverse event; FT4: free thyroxine; FT3: free triiodothyronine; ULN: upper limit of normal; TRAb: TSH-receptor autoantibodies; TPOAb: thyroperoxidase autoantibodies; ATD: antithyroid drug therapy; F: female; M: male; anti-PD-1: anti-programmed cell death 1 protein antibodies; anti-PD-L1: anti-programmed cell death ligand-1 antibodies; anti-CTLA-4: anti-cytotoxic T-lymphocytic-associated antigen-4 antibodies; G3: grade 3 of irAE; DCD: deceased; na: not available. * Corticosteroids were prescribed for other irAE; ** Still on ATD; *** Patient 4 died due to cancer evolution before developing frank hypothyroidism, although the last biochemical examination suggested progression to hypothyroidism (loss of TSH suppression, FT3 and FT4 below the lower limit of normal range).

**Table 2 cancers-13-01944-t002:** ICPis pre-therapeutic Graves’ disease.

No.	Sex Age	Indication	ICPi (Name) and Dose	Total Number of ICPi Cycles	TSH/FT4/FT3 at Time of ICPi Initiation (in ULN)	TRAb at GD’s Diagnosis (in ULN)	TPOAb	Iodine 99mTc Uptake/ Hypervascular Doppler	irAE	ATD Duration (Weeks)	Time to Normalize FT4 (Weeks)	Max TSH/ Min FT4 (in ULN) and (Time)	Time to Normalize TRAb (Weeks)	Status at Last Follow-Up	Treatment at Last Follow-Up (Time of Introduction)	Total Follow-Up (Weeks)
6	M 67	Melanoma	anti-PD1 (Nivolumab) 240 mg/2 weeks	IX	<0.01/1/1	2.4	Negative	High/Yes	No	1	10	13/0.1 (week 16)	na**	Hypothyroidism	Levothyroxine 150 µg/day (week 19)	34
7	M 49	Bronchial adenocarcinoma	anti-PD1 (Pembrolizumab) 200 mg/3 weeks	IX	<0.01/3.4/2.5	1.8	Positive	High/Yes	No	23	8	17/0.5 (week 18)	30	Hypothyroidism	Levothyroxine 250 µg/day (week 21)	94
8	M 74	Renal clear cell carcinoma	anti-PD1 (Nivolumab) 240 mg/2 weeks	XIII	<0.01/1.5/na	14*	Positive	na/Yes	No	16	9	10/0.6 (week 20)	46	Hypothyroidism	Levothyroxine 150 µg/day (week 13)	64

TSH, FT4, FT3 and TRAb are expressed as number of times greater than normal values. Normal values: FT4 12–22 pmol/L, FT3: 3.1–6.8 pmol/L, TSH 0.4–4 mUI/L, TRAb < 1.6 U/L. Time is expressed in weeks since the beginning of thyrotoxicosis (except where otherwise specified). ICPi: immune checkpoint inhibitor; TSH: thyroid stimulating hormone; FT4: free thyroxine; FT3: free triiodothyronine; ULN: upper limit of normal; TRAb: TSH-receptor autoantibodies; GD: Graves’ disease; TPOAb: thyroperoxidase autoantibodies; irAE: immune-related adverse event; ATD: antithyroid drug therapy; M: male; F: female; anti-PD-1: anti-programmed cell death 1 protein antibodies; na: not available. * Patient 3 was diagnosed with GD about 1.5 years prior to ICPi treatment, with ATD for one year; he presented with relapse of GD one month before the initiation of ICPi treatment; ** TRAbs were still positive (3,1 ULN) after 12 weeks of thyrotoxicosis evolution; the patient died before further assays were performed.

**Table 3 cancers-13-01944-t003:** Review of the literature: case reports of Graves’ hyperthyroidism or orbitopathy induced or during ICPi treatment.

First Author, Year	ICPi (Name) and Dose	Sex Age	Personal/Familial History of AID or TD	Hyper-Thyroidism	GO	Cycle/Time	FT4/FT3 (in ULN)	TRAb (ULN)	TPOAb	Iodine 99mTc Uptake	Doppler US Vascular Pattern	Other irAE	Treatment (Duration)	Hypothyroidism Evolution
Graves’ hyperthyroidism														
De Filette, 2016 [19]	anti-PD-1 (Pembrolizumab) 2 mg/kg/na	na	na/na	Yes	No	na	na/na	Positive (na)	Negative	na	na	na	No	Yes
Azmat, 2016 [35]	anti-CTLA-4 (Ipilimumab) 3 mg/kg/2 weeks	M 67	No/na	Yes	No	II/6 weeks	2.1/2.2	Positive (na)	na	High	na	na	ATD (na) then Thyroid-ectomy (a)	Yes (b)
Gan, 2017 [36]	anti-CTLA-4 (Tremelimumab) 1 cycle/12–24 weeks	M 55	No/No	Yes	No	XX/8 years	1.6/1.9	Positive (3.1)	Positive	na	na	na	ATD (na)	No
Brancatella, 2019 [37]	anti-PD-1 (Nivolumab) 3 mg/kg/ 2 weeks	M 51	na/na	Yes	No	IV/8 weeks	1.3/1.6	Negative	Negative	High	Hyper-vascular	na	ATD (na)	No
Iadarola, 2019 [39]	anti-PD-1 (Nivolumab) 3 mg/kg/ 2 weeks	F 66	na/na	Yes	No	II/4 weeks	1/1.3 (c)	Negative	Negative	High	Normal	na	ATD (na)	No
Yajima, 2019 [38]	anti-PD-1 (Pembrolizumab) 200 mg/3 weeks	M 61	No/No	Yes	No	V/14 weeks	2.3/1.7	Positive (2.5) (d)	Negative	na	Hyper-vascular	Colitis	ATD and GC (>20 weeks) (e)	No
Yamada, 2020 [40]	anti-PD-1 (Nivolumab) 240 mg/3 weeks	M 66	No/No	Yes	No	II/6 weeks	>2.9/2.8	Positive (15)	Negative	High	Hyper-vascular	No	ATD (>18 weeks)	No
Kurihara, 2020 [41]	anti-PD-1 (Nivolumab) 240 mg/2–5 weeks	M 48	No/No	Yes	No	VI/16 weeks	1.1/1.2	Positive (1.5)	Negative	Normal	Normal	T1D	ATD (>64 weeks)	No
Graves’ hyperthyroidism and orbitopathy														
Sagiv, 2019 [42]	anti-CTLA-4 (Tremelimumab) 10 mg/kg/4 weeks	M 51	na/na	Yes	Yes	V/27 weeks	>1 (na)	Negative	Positive	na	na	na	GC (na)	No
Graves’ orbitopathy														
Borodic, 2011 [31]	anti-CTLA-4 (Ipilimumab) na	F 51	na/na	No	Yes	II/6 weeks	1/na	Positive (29)	Positive	na	na	na	GC (na)	No
Min, 2011 [30]	anti-CTLA-4 (Ipilimumab) 10 mg/kg/3 weeks	F 51	No/na	No	Yes	IV/12 weeks	1/na	Positive (1.1) (f)	Positive	na	na	na	GC (na)	No
McElnea, 2014 [29]	anti-CTLA-4 (Ipilimumab) 3 mg/kg/2 weeks	F 68	No/No	No	Yes	III/6 weeks	1/na	Negative	Negative	na	na	na	GC (na)	No
Park, 2018 [43]	anti-PD-1 (Pembrolizumab) 1 cycle/2 weeks	M 52	No/Yes (g)	No	Yes	III/9 weeks	na/na	na	na	na	na	na	GC (na)	No
Campredon, 2018 [44]	anti-PD-1 (Nivolumab) na	M 61	No/No	No	Yes	III/6 weeks	1/1	Negative	Negative	na	na	na	GC (na)	No
Sagiv, 2019 [42]	anti-CTLA-4 and anti-PD-1 (Ipilimumab and Nivolumab) 3 mg/kg/2 weeks	M 73	No	No	Yes	II/6 weeks	1/1	Negative	Negative	na	na	na	GC (na)	No
Graves’ disease before ICPis														
Sagiv, 2019 [42]	anti-PD-1 (Nivolumab) 3 mg/kg/2 weeks	M 42	No	Yes (h)	Yes (i)	III/8 weeks	>1 (na)	Positive (na)	Neg/na	na	na	na	No (j)	Yes (k)

FT3 and TRAb are expressed as number of times greater than normal. ICPi: immune checkpoint inhibitor; AID: auto-immune disease; TD: thyroid dysfunction; GO: Graves’ orbitopathy; FT4: free thyroxine; FT3: free triiodothyronine; TSH: thyroid stimulating hormone; TRAb: TSH-receptor autoantibodies; TPOAb: thyroperoxidase autoantibodies; irAE: immune-related adverse event; anti-CTLA-4: anti-cytotoxic T-lymphocytic-associated antigen-4 antibodies; anti-PD-1: anti-programmed cell death 1 protein antibodies; M: male; F: female; ATD: antithyroid drug; GC: glucocorticoids; T1D: type 1 diabetes; ULN upper limit normal; na: not available. (a) Thyroidectomy was performed during surgery to ablate adjacent cancer; (b) Post-surgery; (c) Subclinical hyperthyroidism; (d) TRAb-negative after 7 months of ICPi treatment; (e) Glucocorticoids for colitis; (f) After 17 months of treatment (not available initially); (g) Two cases of hypothyroidism; (h) Diagnosis of Graves’ disease without GO after Pazopanib treatment. Normalized TSH with ATD treatment but TRAb still positive. Nivolumab started 1 year after, with appearance of hyperthyroidism and GO; (i) Clinical Activity Score: 1/10 but other Thyroid Eye Disease signs; (j) Due to lack of ophthalmic inflammatory signs and symptoms; (k) TRAb still positive, whereas secondary hypothyroidism.

## Data Availability

The data presented in this study are available in the manuscript. Complementary data are available on request from the corresponding author.

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
