# Peer review of "Graves’ Disease during Immune Checkpoint Inhibitor Therapy (A Case Series and Literature Review)"

_cancers, 2021, doi:10.3390/cancers13081944_

Round 1

Reviewer 1 Report

In this manuscript Peiffert et. al, presented the status of Immune checkpoint inhibitor (ICPi)-induced thyroid dysfunction leads to Graves disease (GD) through latest scientific literature search and their own case study. They basically talked about the the characteristics and evolution of GD during ICPi therapy. They used various diagnostic tools in their methodology for case study and further data analyzed. The data analysis and their documentation are presented well in this manuscript. They categorized well the disease condition in each patient. The only flaws I observed that writing needs to be more coherent way with their case study and corelated with the latest literature, while reading some of the places I found very complex presentation.  I would also suggest if possible, simplify the data more so the it will be more appealing to scientific community.

Author Response

Point 1:

Writing needs to be more coherent way with their case study and corelated with the latest literature, while reading some of the places I found very complex presentation.  I would also suggest if possible, simplify the data more so the it will be more appealing to scientific community.

Response: 

Respected reviewers,

Thank you for your review of our article “Graves’ Disease during Immune Checkpoint Inhibitors therapy (a case series and literature review)”.

Based on your comments, we have simplified the discussion of the article and reviewed its articulation with the literature.

We have also proposed a graphical abstract summarizing the evolution of each patient category.

Reviewer 2 Report

The manuscript by Prof Peiffert et al deals with the association between exposure to new generation anticancer drugs and adverse reactions affecting the thyroid gland. The authors present their work in the form of a case-series combined with a review of the literature.

The article refers to an uncommon ADR, however, given the presumed increasing use of these therapies, even a rare adverse reaction is of clinical relevance.

I have no particular objections to the article other than the excessive length of the discussion with some aspects of the GD that could be omitted.

Author Response

Point 1: the excessive length of the discussion with some aspects of the GD that could be omitted.

Response 1: 

Respected reviewers,

Thank you for your review of our article “Graves’ Disease during Immune Checkpoint Inhibitors therapy (a case series and literature review)”.

Based on your comments, we have simplified the discussion of the article and reviewed its articulation with the literature. We have removed some details about the GD.

We have also proposed a graphical abstract summarizing the evolution of each patient category.
